# Grief Intervention in Suicide Loss Survivors through Cognitive-Behavioral Therapy: A Systematic Review

**DOI:** 10.3390/bs14090791

**Published:** 2024-09-09

**Authors:** José Carlos Romero-Moreno, María Cantero-García, Ana Huertes-del Arco, Eva Izquierdo-Sotorrío, María Rueda-Extremera, Jesús González-Moreno

**Affiliations:** 1Psicología, Facultad de Ciencias de la Salud, Universidad Internacional de Valencia (VIU), 46003 Valencia, Spain; josecarlosromeromoreno@gmail.com (J.C.R.-M.); jesus.gonzalezm@professor.universidadviu.com (J.G.-M.); 2Psicología, Facultad de Ciencias de la Salud y Educación, Universidad a Distancia de Madrid (UDIMA), 28400 Madrid, Spain; ana.huertes@udima.es (A.H.-d.A.); eva.izquierdo@udima.es (E.I.-S.); maria.rueda@udima.es (M.R.-E.); 3Facultad de Psicología, Universidad Nacional de Educación a Distancia (UNED), 28040 Madrid, Spain

**Keywords:** survivor of suicide, grief, depression, cognitive-behavioral therapy, systematic review

## Abstract

Suicide is one of the leading causes of death worldwide, affecting numerous individuals close to the person who died by suicide, including family members, friends, and colleagues. Those affected by the suicide of someone are referred to as “suicide survivors”, and the psychological consequences they face are particularly severe. One of these consequences is grief, which is more complicated in survivors of suicide compared with those grieving deaths from other causes, mainly because of the stigma that continues to surround them. Therefore, psychotherapeutic intervention for suicide loss survivors is crucial. This study examines the efficacy of cognitive-behavioral programs for addressing grief and other related variables in suicide loss survivors. The search was conducted in databases including Psycinfo, Academic Search Premier, Medline, APA PsycArticles, E-Journals, Scopus, and PubMed. Five randomized controlled trials were selected, one of which focused exclusively on minors. The results reveal that cognitive-behavioral therapy is effective, although the methodological quality of these studies is not adequate, and the representativeness of the samples is very low. More RCTs are needed on the application of cognitive-behavioral programs to treat grief in suicide loss survivors.

## 1. Introduction

Every 40 s, a suicide occurs globally, totaling over 700,000 completed suicides annually [1,2,3]. In the United States alone, suicide rates increased by approximately 36% between 2000 and 2022, with 49,476 deaths in 2022—equating to about one death every 11 min [4]. Each suicide affects an estimated 60 individuals, including family, friends, and colleagues, known as “suicide survivors” [5]. The grief experienced by suicide loss survivors is often more complex because of the stigma surrounding suicide, impacting not only those directly affected by the attempts but also their loved ones, ultimately leading to complicated grief [6,7,8].

Previous reviews have highlighted the cognitive-behavioral model by Boelen et al. [9] as a key theoretical framework for addressing complicated grief. Although these reviews noted a scarcity of studies examining the effectiveness of cognitive-behavioral interventions (CBT) through randomized controlled trials, these reviews are outdated. Therefore, new reviews are necessary to identify recent randomized controlled trials employing cognitive-behavioral therapies in suicide loss survivors to alleviate grief and assess the effectiveness of such therapies in reducing grief in this population.

Suicide can be defined as any self-inflicted behavior performed with the intention of causing oneself intentional death [10,11]. The World Health Organization [3] defines it as an “act with a fatal outcome, deliberately initiated and performed by the deceased in the knowledge or expectation of a fatal outcome through which changes he or she desired to achieve”. According to global data, suicide claims the lives of more than 700,000 people annually [1,2,3], with one suicide attempt every three seconds [3], making it one of the top ten leading causes of non-natural deaths in Europe, America, and much of Asia [2]. These figures have increased because of the situation caused by the COVID-19 pandemic [12].

The etiology of suicide is multifactorial, resulting from the combination of biological, psychological, and social factors that interact differentially according to the individual and culture [13,14,15]. However, studies that seek to establish a “general risk profile” for suicide often focus on middle-aged Caucasian populations, making this profile not globally representative [13]. Additionally, many suicides are not classified as such when “passive” methods are used, such as overdoses or traffic collisions, which are often mistaken for accidental events [8].

### The Impact of Suicide: Addressing Complicated Grief in Survivors

It has been noted that a single suicide typically affects at least six closely related individuals [8,16]; however, these figures could be higher [7], potentially reaching up to 40, including family members and other acquaintances [17]. Based on this, the term “suicide survivor” has become popular, referring to someone close to a person who has died by suicide, attempting to “understand the reasons for the death and learn to continue with their life” [5]. This definition emphasizes the change that the suicide of a loved one brings to the survivor’s life [6]. Despite being a term frequently used by researchers, it also has certain limitations, including confusion about who the term refers to (whether it refers to those who survive the suicide or an individual who attempts suicide and survives) and the difficulty in delimiting the social network to which this term may refer (family members, friends, someone who indirectly collaborates in the suicidal act such as a train conductor when someone jumps onto the tracks to death by suicide, etc.) [6,7]. Based on the conception of this definition, Shneidman [16] used the term “postvention” to refer to all activities and/or interventions carried out to address the psychological consequences arising from the suicide of a loved one.

The death of a loved one is a universally stressful event, which favors the appearance of a set of unpleasant emotional, cognitive, behavioral, and physiological symptoms that can contribute to the development of subsequent psychological problems [18]. It has been estimated that between 6 and 15% of people who experience grief over the loss of a loved one may develop greater complications such as complicated grief [19]. However, in the case of suicide, this probability increases considerably [6,7,8]. For example, a higher risk of suicidal behaviors and a high prevalence of complicated grief have been observed in suicide loss survivors, with feelings of guilt, rumination on the reason for the loved one’s suicide, post-traumatic symptoms, and anger being prominent in this type of grief [7,8]. Complicated grief is defined as a prolongation over time of the normal grief reaction, where feelings of concern, sadness, and emotional pain persist, and positive memories of the deceased are blocked, hindering the recovery of normal life [8]. Although the presence of complicated grief is quite common in suicide loss survivors, in some cases, such as when the deceased was diagnosed with a psychiatric disorder, the process is more manageable, and understanding of the suicidal act by their loved ones is facilitated [6]. Additionally, as noted by Ali [6] and Jordan [7], grief after suicide is experienced in solitude; suicide is still highly stigmatized, so survivors are often blamed for not having done enough to save the individual, leading to grief that is experienced in isolation and shame, even hiding what happened to avoid feeling more responsible than the survivor already feels. Complicated grief in suicide loss survivors is greater the closer the relationship with the deceased; Mitchell et al. [8] observed that the prevalence of complicated grief in suicide loss survivors was twice as high in those who were closely related to the suicide victim compared with those who had a less intimate relationship with them.

In addition to grief, post-traumatic symptoms, guilt, stigma, and depressive and anxious symptoms are also frequent [6]. Hopelessness and yearning, characteristics present in depressive episodes, are very common in complicated grief, which may be related to suicidal ideation and behaviors with the intention of avoiding suffering and reuniting with the deceased [8], explaining the elevated risk of suicide in suicide loss survivors.

Boelen et al. [9] describe the development and maintenance of complicated grief from a cognitive-behavioral perspective, highlighting the following three key phases in this process: (a) Poor elaboration and integration of the death into the subject’s autobiographical information: Individuals with complicated grief, despite being exposed to numerous stimuli reminding them of the deceased, are unable to assimilate the loss, continuing to perceive it as “unreal”. (b) Global and negative beliefs about oneself, life, and the future and erroneous interpretations of grief: The loss of a loved one and the role they played in the mourner’s life can disrupt the stable and global beliefs the mourner had about life and the future. As Boelen et al. [9] point out, the death of a child can affect beliefs about the meaning of life, while the death of a romantic partner can alter expectations for the future. This disruption of “future-directed stability” can affect daily functioning. Furthermore, the loss may not only affect future expectations but also reactivate negative beliefs about oneself that the presence of the deceased had previously mitigated. The authors provide an example of how the death of a romantic partner can trigger latent negative beliefs that were suppressed while the partner was alive. These beliefs, combined with the interpretation that the loved one’s death is an unbearable pain and intrusive memories of the deceased (erroneous interpretations of grief), lead to avoidance, which in turn sustains complicated grief. Guilt and anger are also central to this process, especially in cases of suicide grief, where the mourner may feel responsible for the death or struggle to understand why it happened. (c) Avoidance of grief-related reactions: These strategies include anxious and depressive behaviors, such as avoiding thoughts, memories, and feelings related to the deceased in order to prevent “going crazy”.

Examples of these are avoiding exposure to places the mourner visited with the deceased, to people who may talk about the deceased, or to objects that remind them of the deceased (although, sometimes the opposite effect occurs, where avoidance occurs towards “no contact” with those objects, people, or situations, with the intention of avoiding contact with a reality without the deceased), and rumination on the reason for the loss. Among the latter, depressive strategies stand out, including those in which the mourner enters a state of hypoactivity, ceasing to participate in social events and pleasurable activities; in this case, the goal is not to avoid stimuli, but rather the cause is the lack of reinforcement or guilt for “enjoying without the deceased”.

The authors include in this model other variables such as individual vulnerability factors, event characteristics, and the consequences of loss, along with another set of variables and processes that make up the Boelen et al. [9] model. Based on the cognitive-behavioral model of complicated grief developed by Boelen et al. [9], cognitive-behavioral therapy has frequently been applied to treat complicated grief, finding positive effects of this intervention [20]. However, fewer studies have analyzed, with adequate methodological quality, the efficacy of cognitive-behavioral therapy and other types of interventions in cases of complicated grief due to suicide [21].

In 2008, Jordan [7] noted the lack of empirical support with controlled designs on interventions for grief in suicide loss survivors while highlighting that individual therapy, joint family therapy, and support groups appear to be useful for these survivors. Similarly, considering the high prevalence of post-traumatic symptoms in these individuals, some behavioral interventions such as exposure therapy, techniques like empty chair or narrative therapy, and interventions such as eye movement desensitization and reprocessing (EMDR) could be useful [7]. Linde et al. [1] conducted a systematic review of interventions used in grief among suicide loss survivors, finding that cognitive-behavioral interventions were effective in suicide loss survivors with high suicidal ideation, although only two cognitive-behavioral programs were identified in their review. In a systematic review conducted by Andriessen et al. [22], promising results were also found for cognitive-behavioral programs, although the review was conducted only a year after Linde et al.’s study [1], so they did not find many more publications regarding this type of program because of the short time lapse between one review and the other.

Taking into consideration that the most recent review available was conducted by Andriessen et al. [22] and was performed in 2018, it is important to conduct new systematic reviews to determine if there are more recent studies applying cognitive-behavioral programs in grief among suicide loss survivors, as well as the efficacy of such programs. Based on the PICO questions, the general objective of this work is to determine the efficacy of cognitive-behavioral programs (intervention) compared to control groups (comparison) in reducing grief (outcome) in suicide loss survivors (patients). The specific objectives of this systematic review are as follows: 1. Determine how many RCTs have been conducted to analyze the efficacy of cognitive behavioral interventions in reducing complicated grief in suicide loss survivors. 2. Identify the types of relatives to whom these cognitive-behavioral programs have been applied. 3. Determine the types of cognitive behavioral programs used in the reviewed RCTs (format, duration, and components). 4. Identify the types of controls used in the reviewed RCTs. 5. Determine if cognitive-behavioral programs are effective in reducing grief related to suicide. 

## 2. Materials and Methods

This systematic review was conducted according to PRISMA criteria [23]. The search process began on 1 April and ended on 20 April 2024, using Psycinfo, Academic Search Premier, Medline, APA PSycArticles, E-Journals, Scopus, and Pubmed databases. The following search equation was used in all databases: (cognitive-behavioral therapy OR cognitive-behaviour therapy) AND (grief OR mourning OR bereavement OR depression) AND suicide AND intervention AND controlled. No temporal restrictions were established regarding publications. RCTs were accepted only if they included suicide loss survivors in grief as participants and if their primary measure was related to grief (depression, suicidal ideation, post-traumatic symptoms, rumination, and/or guilt). Samples of both adult and child participants were accepted, and survivors with both familial and non-familial close relationships with the deceased were admitted. Studies using intervention programs that, while not exclusively cognitive behavioral, were largely based on this theoretical framework were also accepted. Qualitative studies, systematic reviews, meta-analyses (although consulted for new records), single case studies, books, book chapters, studies not focusing on survivors, and the gray literature were excluded from this review. Studies published in languages other than English and Spanish were also excluded. After performing all the aforementioned searches, the obtained records were identified, and duplicate records were removed. Subsequently, the first screening was conducted based on the title and abstract of each identified record. Studies that did not meet the criteria were eliminated after this screening and, in the second screening, each record was read in its entirety to eliminate records that did not meet the criteria or were inaccessible. The selection of relevant studies involved a three-phase process carried out by two reviewers (R.M.J.C. and M.C.G.). First, the titles and abstracts of all identified papers were assessed according to the pre-established inclusion and exclusion criteria. In the second phase, the full texts of papers deemed potentially relevant were examined. This phase concluded with a cross-reference check.

Data analysis was performed using qualitative synthesis, and the risk of bias of each accepted study in the review was analyzed based on Cochrane criteria [24] aimed at evaluating the risk of selection, performance, detection, attrition, and reporting bias. 

The authors registered this manuscript in Open Science Framework (OSF) at 24 July 2024.

## 3. Results

During the search process, a total of 405 records were identified, with 109 identified in the Scopus database, 99 in Psycinfo, 93 in Medline, 65 in Academic Search Premier, 29 in E-Journals, 7 in Pubmed, and 3 in APA Psycarticles. Additionally, an additional record was identified from the previous reviews. Of these 406 records, 166 were duplicates, resulting in 240 unique records after removing duplicates. In the first screening, titles and abstracts of each record were analyzed, leading to the elimination of 235 studies that did not meet the criteria (219 of them either addressed grief but not suicide or addressed suicide but not survivor grief, 7 were responses to other authors, editorials, or conference posters, 6 were not in English or Spanish, 2 were systematic reviews, and 1 was a protocol for intervention for survivor grief but had not yet been implemented). Thus, five records passed the first screening and were read in their entirety in the second screening, and they were accepted for meeting all selection criteria, with four of them coming from database searches and one being an additional record. In Figure 1, the flowchart is shown.

A qualitative analysis of the accepted studies was conducted, and the data obtained were synthesized based on the following four themes: study design, description of interventions, assessment instruments used, and results obtained. Table 1 presents the synthesis of these data.

### 3.1. Methodological Quality Assessment

A study was conducted to assess the methodological quality of the reviewed studies. Reporting bias was high in all studies, as significance data were frequently not provided for non-significant differences or effect sizes were not reported. Only the study by Treml et al. [29] provided information on the method of handling dropouts during the intervention. All studies, except for Pfeffer et al. [25], reported the method of randomization used. However, there were no guarantees regarding the blinding of participants and personnel in most studies. Pfeffer et al. [25] had the lowest methodological quality as it did not provide data on any of the analyzed criteria. In Figure 2 and Figure 3, the risk of bias graph and the summary risk of bias graph, respectively, of the reviewed RCTs are shown.

### 3.2. Participant Description

The mean age of the total sample obtained in the review was 36.67 years (SD = 11.08), with a total of 350 suicide loss survivors, of which 27.43% were male. All studies except for Pfeffer et al. [25] included adult survivors, although DeGroot et al. [26] and De Groot et al. [27] allowed participation from individuals aged 15 and above. Participants in De Groot et al. [26,27] were mainly spouses (36 participants), parents (29 participants), children (27 participants), and siblings (21 participants). In Wittouck et al. [28], most participants were children of the deceased (33 participants), spouses (20 participants), siblings (15 participants), and parents (8 participants). In Treml et al. [29], participants were primarily parents (19 participants), children (11 participants), siblings (11 participants), and spouses (10 participants). All studies accepted other participants with a close relationship with the deceased (friends, etc.). Finally, in Pfeffer et al. [25], participants were children (46 participants) and siblings of the deceased (25 participants).

### 3.3. Description of Interventions

Only participants in the experimental conditions of each study received an intervention, while participants in the control conditions were placed on a waiting list. Notably, the study by De Groot et al. [27] is an extension of the study by De Groot et al. [26], analyzing differences in the effectiveness of the intervention between grieving survivors with and without suicidal ideation. Although all intervention programs were based on the cognitive-behavioral model, there was a great variety in the application format and structure of each program.

The interventions described in De Groot et al. [26,27] and Wittouck et al. [28] were conducted through home visits. Treml et al. [29] utilized an online format, while Pfeffer et al. [25] did not specify the location of the intervention. De Groot et al. [26,27] and Wittouck et al. [28] were based on the cognitive-behavioral model of Boelen et al. [9]. In De Groot et al.’s studies [26,27], the intervention involved psychotherapeutic counseling conducted by psychiatric nurses who had received prior training in this cognitive-behavioral intervention. Each family was assigned a specialized nurse who supervised each session conducted in the family’s home. The intervention aimed to involve the entire family system, providing psychoeducation, emotional processing, effective interaction techniques, and problem-solving techniques. The program consisted of fixed modules (cognitive restructuring and support consolidation) and optional modules tailored to each family’s needs. Wittouck et al. [28] mainly provided psychoeducation on suicide and discussions on various grief myths. Treml et al. [29] implemented an online narrative therapy program based on Lange et al. [30] and adapted by Wagner et al. [20] for individuals experiencing complicated grief.

Pfeffer et al. [25] used an eclectic program based on various theories and models, including Bowlby’s attachment theory [31], Ness and Pfeffer’s loss response theory [32], and Lazarus and Folkman’s cognitive coping theory [33]. The program consisted of the following main components: psychoeducation and support, aimed at addressing the understanding of death and suicide, and the loss of psychosocial and personal resources.

### 3.4. Intervention Effectiveness

Most interventions were effective in reducing grief, both in direct grief measures and related symptoms (depressive, anxious, post-traumatic, or suicidal). While there was a non-significant reduction in maladaptive grief reactions, it was more pronounced in survivors with suicidal ideation. Significant reductions were observed in maladaptive grief symptoms post-intervention in Wittouck et al. [28] and in the severity of grief symptoms in Treml et al. [29], which persisted over time. Depressive symptom reduction was significant in all studies except for De Groot et al. [26,27]. Only Pfeffer et al. [25] included anxious symptomatology, showing a significant reduction after intervention. Post-traumatic symptomatology was included only in Pfeffer et al. [25], where no significant changes were found after intervention. Regarding suicidal symptomatology, no significant changes were found in the studies by De Groot et al. [26,27], although the intervention was more effective in those with suicidal ideation [27].

## 4. Discussion

This work aimed to systematically review randomized controlled trials using cognitive-behavioral therapy (CBT) interventions to alleviate grief and related symptoms in suicide loss survivors. It found a limited number of such trials, with generally low methodological quality. Despite the high risk of complications among suicide loss survivors, there is a lack of studies applying CBT, highlighting a gap in intervention research. While there is some evidence suggesting the effectiveness of CBT in alleviating grief-related symptoms, the mixed results across studies make it challenging to draw definitive conclusions.

It is crucial to recognize the importance of designing and implementing CBT interventions that are tailored to the specific characteristics of both younger and older adults. While most research tends to focus on younger populations, particularly because of concerns over rising suicide rates in this group, it is equally vital to consider older adults. This group often faces multiple risk factors, such as social isolation, the loss of loved ones, and the presence of chronic illnesses, and may benefit from interventions that specifically address their unique circumstances and challenges.

Adapting interventions to each age group would not only better meet their needs but could also enhance the effectiveness of the therapies applied. For example, younger individuals may require approaches that take into account emotional development and the influence of social networks, whereas older adults might benefit more from strategies that integrate social support and address loneliness and accumulated grief throughout life. Considering these differences in the design and implementation of future studies could provide more precise and applicable data for each demographic group.

These findings agree with Wilson and Marshall [34], who noted the insufficient availability and quality of psychological support services for grieving suicide loss survivors. The impact of cognitive-behavioral therapy on grief and related variables such as anxiety, depression, and post-traumatic symptoms shows mixed results in the literature. While some studies have reported significant reductions in grief symptoms, others have not, and the authors did not offer explanations for these discrepancies. Additionally, some studies only provided partial results without reporting effect sizes, and missing data were not addressed using specific analytical methods. Moreover, it was observed that suicidal ideation among suicide loss survivors influenced the interventions’ effectiveness, with greater efficacy observed among individuals with suicidal ideation. However, this finding may be biased since individuals with higher suicidal ideation also tended to have higher levels of anxiety, depression, and neuroticism, suggesting a greater potential for improvement in these individuals.

In Pfeffer et al. [25], although no specific grief measure was included, significant reductions were found in depression and anxiety symptoms related to the suicide of parents or siblings, but not in post-traumatic symptoms. The authors suggested that the lack of an effect on post-traumatic symptoms could be due to low social adaptation, which might improve over time, leading to a reduction in post-traumatic symptoms. However, the authors noted potential biases due to the small sample size and high experimental mortality rate (i.e., loss of participants during the course of an experiment or study). They did not offer explanations for why the intervention seemed more effective in middle-aged children. In Wittouck et al. [28], although reductions in depressive symptoms and grief-related symptoms were observed post-intervention, there were no reductions in hopelessness or negative cognitions related to grief distress. The passive coping style (worrying, rumination, pessimism, etc.) decreased, which is crucial in the cognitive-behavioral model of complicated grief by Boelen et al. [9]. However, participants in the experimental group decreased their seeking of social support post-intervention, possibly because of the emotional support experienced during the intervention. The authors also noted a low sample size and potential selection bias among participants. In Treml et al. [29], positive results were found in grief symptoms, depression, guilt, and shame measures, common issues among suicide loss survivors because of the stigma associated with suicide, which interferes with normalization and adaptation. However, general psychopathology improved in both the experimental and control groups. The authors attributed these results to the online intervention’s acceptance among users, suggesting that the protection of user data may reduce fear of judgment, encouraging more expression. However, the authors cautioned that this could increase the likelihood of misunderstandings. Despite the positive intervention results, caution is warranted because of the limited generalizability, as most of the sample were highly educated women living with a partner, which facilitates social support, a crucial variable in grief adaptation according to Boelen et al. [9]. Participants with various psychopathologies, including suicidal ideation, were excluded, potentially biasing the results.

This review has limitations, including the selection of only randomized controlled trials published in scientific journals, potentially reducing the number of studies obtained. Additionally, because of the variety of variables collected in each study, a meta-analysis was not performed, limiting specificity regarding intervention effectiveness. Therefore, while our findings suggest that CBT can be effective in certain contexts, more randomized controlled trials are needed to establish its full efficacy and to explore other potential strategies for supporting suicide loss survivors. The lack of a clinically significant cutoff point for complicated grief diagnosis and consideration of the time frame needed to classify grief as a mental disorder may have affected the observed reductions in grief symptoms. Future reviews should include more grief-related variables and focus on tertiary interventions targeting individuals clinically diagnosed with complicated grief. It is also essential to analyze intervention effectiveness by age, as studies specific to older adults were lacking.

While there is some evidence indicating that cognitive-behavioral therapy may be helpful in reducing grief symptoms in suicide loss survivors, along with other related symptoms, the methodological quality of the studies, small sample sizes, low representativeness, and high experimental mortality rate make it challenging to draw general conclusions about intervention effectiveness. More randomized controlled trials analyzing the efficacy of cognitive-behavioral therapy in suicide loss survivors, both in adults and children, with more representative samples, are needed.

Future research should focus on conducting higher-quality randomized controlled trials with larger and more diverse samples, including participants with various psychological profiles, such as those experiencing suicidal ideation, to improve the generalizability of the findings. It is also important to include diverse demographic groups, such as older adults, ethnic minorities, individuals from various socioeconomic backgrounds, and men, to better evaluate the generalizability of interventions for suicide loss survivors. Additionally, there is a critical need to develop and validate standardized measures for assessing grief, particularly in the context of suicide bereavement, which would enable more consistent comparisons of outcomes across studies. Investigating the long-term effectiveness of cognitive-behavioral therapy (CBT) interventions through follow-up assessments at multiple time points is recommended to determine whether the benefits are sustained or if additional interventions are required. Moreover, future research should compare the effectiveness of CBT with other psychological therapies, such as Acceptance and Commitment Therapy (ACT), Narrative Therapy, Dialectical Behavior Therapy (DBT), Client-Centered Therapy (CCT), or Brief Psychodynamic Psychotherapy (BPP). In addition to RCTs, non-RCT studies, such as large-scale cohort studies, real-world effectiveness trials, and case-control studies, should also be considered to explore the effectiveness of these interventions in practical, real-world settings. For example, cohort studies could track outcomes over time in naturalistic settings where randomization is not feasible, providing valuable insights into how these therapies perform outside controlled experimental conditions. Real-world effectiveness trials could assess the impact of these interventions across diverse healthcare settings, including community clinics and telehealth platforms, where conditions are less controlled than in traditional RCTs. Additionally, case-control studies could be used to investigate the effectiveness of interventions in specific subgroups, such as individuals with concurrent mental health disorders or those from underserved populations. This comparison would help identify the most effective strategies for different groups of suicide loss survivors. Furthermore, it is essential to delve deeper into the role of social support in grief adaptation, especially within online intervention contexts, and to establish clinically significant cutoff points for the diagnosis of complicated grief, taking into account the necessary time frame for classifying grief as a mental disorder. Future research should also explore the specific mechanisms through which these various therapies reduce grief and related symptoms to enhance the development of more targeted clinical interventions.

## Figures and Tables

**Figure 1 behavsci-14-00791-f001:**
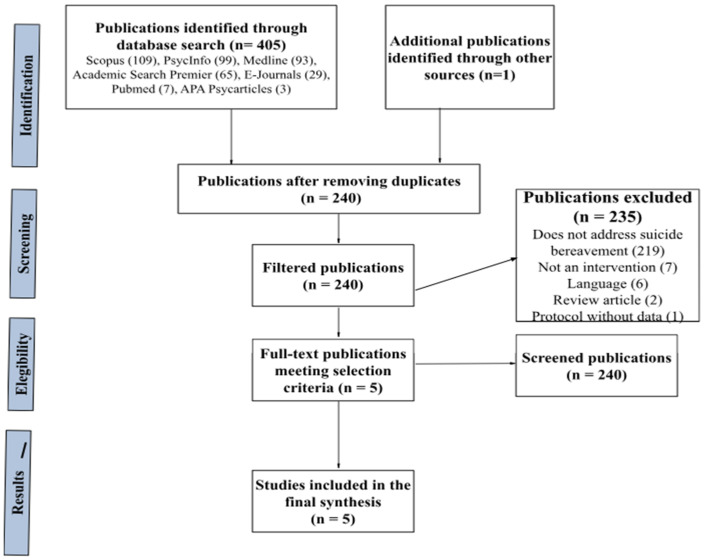
PRISMA-based systematic review process flowchart.

**Figure 2 behavsci-14-00791-f002:**
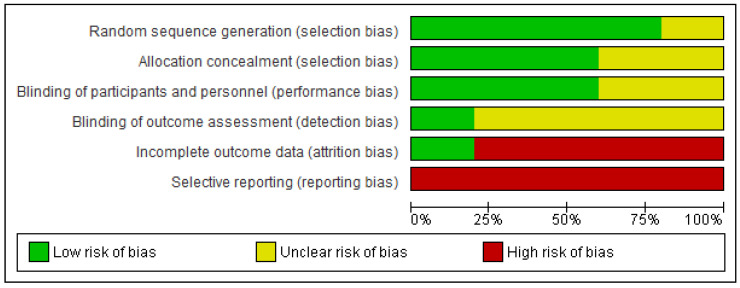
Risk of bias graph.

**Figure 3 behavsci-14-00791-f003:**
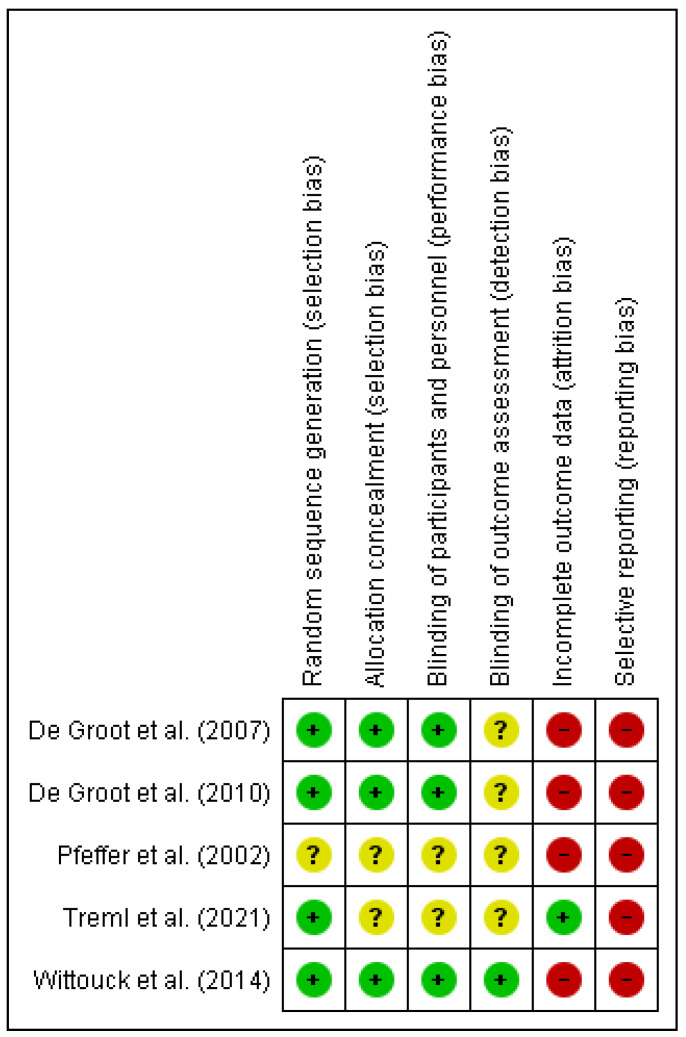
Risk of bias graph (see Figure 2 for explanation of symbols used) [25,26,27,28,29].

**Table 1 behavsci-14-00791-t001:** Summary of results.

Author and Year	Design	Intervention	Measurements	Results
Pfeffer et al. [25]	A total of 75 children (N_experimental_ = 39; N_control_ = 36). Mean age = 10.5 years old, SD = 3.2; 28 men. Pre- and post-intervention measures	Experimental group: BGI × 10 weekly group sessions, 1.5 h per session. Two components: psychoeducation and support. Groups of two–five children, organized by age (6–9 years old; 10–12 years old; 13–15 years old). Control group: Waiting list.	CPTSRI for post-traumatic symptomatology, CDI for depressive symptomatology, RCMAS for anxious symptomatology, SAICA to assess social adjustment.	Between-group comparisons: Significantly greater reductions in anxiety symptoms (*p* < 0.009; d = 0.80) and depressive symptoms (*p* < 0.01; d = 0.70) were observed in the experimental group compared with the control group, with larger reductions observed in middle-aged children (11 years old) compared with younger (6 years old) and older (14 years old) children. No significant differences were found in post-traumatic symptomatology or social adjustment. Within-group comparisons: Significant reductions were observed in post-measure anxiety symptoms (*p* < 0.04) and depressive symptoms (*p* < 0.003) in the experimental group, with a non-significant reduction in post-traumatic symptomatology (*p* < 0.06).
De Groot et al. [26]	A total of 134 participants > 15 years old (N_experimental_ = 74; N_control_ = 60). Mean age = 43 years old, SD= 13.6; 40 men. Preintervention measures (2.5 months after the suicide) and post-intervention measures (13 months after the suicide).	Experimental group: CBT counseling × four sessions of 2 h each, one session every 2–3 weeks. Intervention for the entire family system. Components: Psychoeducation, emotional processing, effective interaction, and problem-solving. Based on the theory of Boelen et al. [9]. Control group: Waiting list.	ITG complicated grief, CESD depressive symptomatology, PSI suicidal ideation, responsibility for suicide feelings with ad hoc questionnaire, TRGR2L maladaptive grief reactions.	Between-group comparisons: No effect on complicated grief (*p* = 0.82), depression levels (*p* = 0.28), or the presence of suicidal ideation, but there was an effect on maladaptive grief reactions with a notable but not significant reduction (*p* = 0.056) and on guilt perception with a significant reduction (*p* = 0.01). No within-group comparisons either each group.
De Groot et al. [27]	A total of 134 participants > 15 years old (N_experimental_ = 74; N_control_ = 60). Mean age= 43 years old, SD= 13.6; 40 men. Preintervention measures (2.5 months after the suicide) and post-intervention measures (13 months after the suicide).	Experimental group: CBT counseling × four sessions of 2 h each, one session every 2–3 weeks. Intervention for the entire family system. Components: Psychoeducation, emotional processing, effective interaction, and problem-solving. Based on the theory of Boelen et al. [9]. Control group: Waiting list.	EPQ-RSS, neuroticism; perceived sense of control over one’s life; self-esteem with RSES; family history of suicide; subjective expectation of suicide index; responsibility for suicide feelings with ad hoc scales; complicated grief with ITG; depressive symptoms with CESD; suicidal ideation with PSI; maladaptive grief reactions with TRGR2L; anxiety and depression with SCAN 2.1	Survivors with suicidal ideation had a higher history of anxiety (*p* < 0.05), depression (*p* < 0.01), suicidal behavior (*p* < 0.001), and neuroticism (*p* < 0.001) and lower self-esteem (*p* < 0.001) compared with survivors without suicidal ideation. No significant differences were found in the interaction between suicidal ideation and intervention factors on the variables of complicated grief (*p* = 0.33), depressive symptoms (*p* = 0.57), or guilt perception (*p* = 0.60). There was a significantly greater reduction in maladaptive grief reactions (*p* = 0.03) and suicidal behavior (*p* = 0.03) among relatives with suicidal ideation
Wittouck et al. [28]	A total of 83 adult participants (N_experimental_ = 47; N_control_ = 36). Mean age = 48.6 years old, SD = 13.3; 20 men. Measures pre- and post-intervention. Four additional visits during the intervention in the experimental group.	Experimental group: CBT × home sessions lasting 2 h each session. Components: Psychoeducation about suicide, grief, specific aspects of grief due to suicide, and coping with grief. Based on the theory of Boelen et al. [9]. Control group: Waiting list.	Maladaptive grief symptomatology with ITG, depressive symptomatology with BDI-II, hopelessness with BHS, negative cognitions related to grief distress with CGQ, maladaptive coping with UCL.	Reducción significativa en el grupo experimental en sintomatología desadaptativa de duelo (*p* = 0.021) y en sintomatología depresiva (*p* = 0.006), pero no en desesperanza (*p* = 0.231). Sin reducción en grupo control ni en sintomatología de duelo (*p* = 0.503), ni en sintomatología depresiva (*p* = 0.250) ni en desesperanza (*p* = 0.688). Sin reducción en ninguna de las dimensiones del GCQ ni en grupo experimental ni en control. Reducción significativa en grupo experimental en las dimensiones del UCL “apoyo social” (*p* = 0.002), “reacción pasiva” (*p* = 0.013) y “expresión emocional” (*p* = 0.004).
Treml et al. [29]	A total of 58 adults (N_experimental_ = 30; N_control_ = 29). Mean age = 44.57 years old, SD= 14.25; 8 men. Measures pre- and post-intervention and follow-up at 3, 6, and 12 months after completing the intervention.	Experimental group: Online CBT × psychoeducation about suicide and suicide grief + 10 writing tasks (narrative therapy). Three phases: Coping, cognitive restructuring, and social exchange. Control group: Waiting list.	Severity of grief symptomatology with ICG, grief reaction after loss by suicide with GEQ, depressive symptomatology with BDI-II, general psychopathology with BSI.	Significant reductions in the follow-up measure at 12 months with respect to the post-measure in the experimental group versus the control in the GEQ subscales “guilt” (*p* = 0.043) and “shame” (*p* = 0.015), without differences among follow-up measures. No differences in follow-up in depressive symptoms or psychopathology in general.

Abbreviation Index: Depression Inventory-II; BHS: Beck Hopelessness Scale; BSI: Brief Symptom Inventory; CESD: Center for Epidemiologic Studies depression scale; CGQ: Grief Cognitions Questionnaire; EPQ-RSS: Eysenck Personality Questionnaire; GEQ: Grief Experience Questionnaire; ICG: Inventory of Complicated Grief; ITG: Inventory of Traumatic Grief; PSI: Paykel’s Suicidality Items; RSES: Rosenberg Self Esteem Scale; SCAN 2.1: Schedules for Clinical Assessment in Neuropsychiatry; TRGR2L: Traumatic Grief Evaluation of Response to Loss; UCL: Utrecht Coping List.

## Data Availability

The original data presented in the study are openly available in OSF at reference [35].

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
