# Peer review of "Grief Intervention in Suicide Loss Survivors through Cognitive-Behavioral Therapy: A Systematic Review"

_behavsci, 2024, doi:10.3390/bs14090791_

Round 1
Reviewer 1 Report
Comments and Suggestions for Authors
This is well written and interesting. The results are somewhat luke-warm, as they call for more research and more RCTs. But the idea that one can (or should) study the sufferers and grievers of relatives and friends from suicide is an important take-away. CBT may not be the only therapeutic needed (or working) to manage the grief of relatives. Can the authors include something about this in their the discussion? Since the systematic review on cognitive behavioral theory didn't cover many articles, and those reviewed were of marginal quality, are there other strategies (i.e. counseling, psychotherapy, desensitization, medication, etc) that the authors suggest for further research into overcoming grief from friends and family who are victims of suicide?
Line 41 Since this is the first time you have introduced CBT, use that (CBT) abbreviation for the first time on line 41, after the name. You do this on line 362 in the Discussion, but that is way too late to introduce the abbreviation.
In your introduction to suicide statistics, you might want to reference the US CDC recent information on suicide and prevention. see below.
-
www.cdc.gov › suicide › factsFacts About Suicide | Suicide Prevention | CDC
cdc suicde prevention training cdc suicide prevention programs
Line 29-108 you may consider reducing the length of this section on background on suicide, and add a sub-head when you begin talking about grief issues and survivors' psychology. (line 110)
line 157-196 This paragraph should be split into several others. ....as it is too long for one paragraph. try to find logical breakpoints.
line 242-252 did one reviewer do all the reviews and selections (inclusion and exclusion) or did a second person verify or agree with the results? How many people did the screening?
line 382 Can you define "experimental mortality rate"
References:
line 440....missing vol #
line 445 misspelled "official"
line 451 is this a journal or book? Vol needed?
line 260. Can you define OSF here?
Line 419-428 I believe these are suggested categories to include about author contributions. They don't all need to be completed...only those that are relevant to your particular research.
Line 411 and 414. You state that evidence suggests effectiveness, then in 414 you say "makes it challenging to draw general conclusions". Either the research study finds effectiveness or it doesn't. Perhaps toning down the "effective" statement such as "within the limitations of our study ( x, y, z,) we found .....however, more randomized controlled trials....and investigating other strategies including non-RTC studies for ......are needed in future research..." include some of this language also in the abstract.
Can you suggest some things future research should tackle? perhaps in a section/paragraph "Future research opportunities".
Author Response
Revisor 1
This is well written and interesting. The results are somewhat luke-warm, as they call for more research and more RCTs. But the idea that one can (or should) study the sufferers and grievers of relatives and friends from suicide is an important take-away. CBT may not be the only therapeutic needed (or working) to manage the grief of relatives. Can the authors include something about this in their the discussion? Since the systematic review on cognitive behavioral theory didn't cover many articles, and those reviewed were of marginal quality, are there other strategies (i.e. counseling, psychotherapy, desensitization, medication, etc) that the authors suggest for further research into overcoming grief from friends and family who are victims of suicide?
Line 41 Since this is the first time you have introduced CBT, use that (CBT) abbreviation for the first time on line 41, after the name. You do this on line 362 in the Discussion, but that is way too late to introduce the abbreviation. Thank you for your comments. The authors have incorporated the suggestions on page 1.
In your introduction to suicide statistics, you might want to reference the US CDC recent information on suicide and prevention. see below.
-
www.cdc.gov › suicide › facts
- Facts About Suicide | Suicide Prevention | CDC
Apr 25, 2024 · Belonging, safety, dignity, and hope can protect against suicide. Suicide rates increased approximately 36% between 2000–2022. Suicide was responsible for 49,476 deaths in 2022, which is about one death every 11 minutes. 2 The number of people who think about or attempt suicide is even higher.
cdc suicde prevention training cdc suicide prevention programs Thank you for your comments. The authors have incorporated the suggestions on page 1.Pg 1)
Line 29-108 you may consider reducing the length of this section on background on suicide, and add a sub-head when you begin talking about grief issues and survivors' psychology. (line 110) Thank you for your comments. The authors have incorporated the suggestions.
line 157-196 This paragraph should be split into several others. ....as it is too long for one paragraph. try to find logical breakpoints. Thank you for your comments. The authors have incorporated the suggestions on pgs 4 and 5
line 242-252 did one reviewer do all the reviews and selections (inclusion and exclusion) or did a second person verify or agree with the results? How many people did the screening?
Thank you for your comments. The authors have incorporated the suggestions. The selection of relevant studies followed a three-phase process conducted by two reviewers JCMR & MCG : first, the titles and abstracts of all identified papers were reviewed based on the previously established inclusion and exclusion criteria. In the second phase, the full texts of the works identified as potentially relevant were reviewed. This phase was completed with a cross reference check
line 382 Can you define "experimental mortality rate" (Done. L 382)
References: Thank you for your comments. The authors have reviewed all the references and formatted them in Vancouver style. Done
Lline 440....missing vol # Done
line 445 misspelled "official" Done
line 451 is this a journal or book? Vol needed? Done
line 260. Can you define OSF here? Open Science Framework
Line 419-428 I believe these are suggested categories to include about author contributions. They don't all need to be completed...only those that are relevant to your particular research. Done. The authors have made the necessary modifications to the manuscript. They have removed this information to enhance the readability and comprehension of the text.
Line 411 and 414. You state that evidence suggests effectiveness, then in 414 you say "makes it challenging to draw general conclusions". Either the research study finds effectiveness or it doesn't. Perhaps toning down the "effective" statement such as "within the limitations of our study ( x, y, z,) we found .....however, more randomized controlled trials....and investigating other strategies including non-RTC studies for ......are needed in future research..." include some of this language also in the abstract. The authors have made the necessary modifications to the manuscript.The authors have reviewed the entire discussion section.
Can you suggest some things future research should tackle? perhaps in a section/paragraph "Future research opportunities". Thank you for considering this. The authors have included future research directions.P. 11
Reviewer 2 Report
Comments and Suggestions for Authors
This manuscript presents an overview of grief among suicide loss survivors, with a focus on cognitive behavioral aspects and approaches to suicide grief experiences. The goal of the article is to analyses the research literature on suicide grief, identifying research investigations that have utilized randomized control trials and evaluating the status of the quality and findings of cognitive behavioral therapeutic (CBT) approaches for effectiveness after suicide loss. After extensive search for such research findings, 5 investigations were identified and evaluated. Based on these studies, the authors observed that CBT are effective while the quality of the studies was inadequate and there are limitations to the studies and their methods.
Strengths of the study lie in the number of databases included in the search for identifying research on CBT for suicide grief and mourning to include in the review of effectiveness. The use of the PRISMA criteria in identifying the studies is another strength.
There are some issues and questions that might be addressed.
(1) The entire paragraph for Section 3.1 Methodological Quality Assessment (lines 295-306) is an exact repetition of the paragraph of 3. Results (lines 265-277). No new information appears in Section 3.1’s text and more important, no information about Methodological Quality Assessment is included, and no information about Figures 2 and 3 appear.
(2) The sentence in the introduction, lines 106-107 ends midsentence,. Table 1 is not mentioned in the text and an ending is needed in the sentence. Was there more to add to the Halabi and Garcia table of factors to better understand the inclusion of Table 1?
(3) For Section 3.2 Participant Description, the heterogeneity of the participant relationships is a plus and minus. The plus is a better representation of some of the variety of possible relationships between the deceased and survivors. A minus is that little is known about whether these interventions are effective for some relationship groups but less so or not for others.
(4) On line 113 the “popular” use of the term “suicide survivor” is discussed, along with its potential confusion with reference to suicide attempters. However, for some time now the writing and research on the topic of suicide survivors has utilized “suicide loss survivors” for the previously utilized “suicide survivors” terminology alone. At the same time, suicide attempters have often been referred to as “suicide attempt survivors.” The mention of the previous terminology and the confusion it potentially represented is accurate, but at this point in time the issue is minimal and essentially addressed in the language utilized.
(5) Included in the review in the introduction, the work of Edwin Shneidman (though a secondary source is cited) appears on line 123. It should be noted that Shneidman’s name does not include a “c” (i.e., “Schneidman) as appears in line 123.
(6) (a) On line 122 the wording “commit suicide” appears. While this wording has been used in the research and clinical literature in the past (and unfortunately sometimes still is), current practices and recommendations strongly encourage avoidance of such phrasing in labeling deaths by suicide. Instead, wording such as died by suicide, die by suicide, died of suicide, killed themselves, suicide death, death by suicide, or some other wording is suggested. The term “commit suicide” implies, suggests, or is associated with a criminal or immoral act. It is interpreted as a pejorative word and one that is harmful to the surviving family of those who have died by suicide. That implication is associated with feelings of shame and stigmatization toward the individual and often their surviving family members as well.
(7) In the Material and Methods section (lines 247 ff) it is noted that a first and second screening were conducted. Who conducted the screenings is not identified here or in the Author Contributions listing at the end of the manuscript. Were the evaluations conducted by all authors, or a subset of authors, research assistants or some other persons?
(8) The authors contend that the passage of sufficient time has taken place to justify a review of this research topic. The last review is stated to have taken place in 2018. When considering the 5 studies identified and reviewed (no time frame was placed on the search period), only one of the studies (Treml et al., 2021, apparently the “additional record” identified - or was it identified in the searches but another of the other four, though appearing before 2018, represents the “additional record?) was new and identified for inclusion. While this may be the only new study meeting the established criteria, the significance or implication of the lack of other research on this topic after a 6-year period of time is not addressed by the authors. Therefore, while seemingly positive results of effectiveness are stated, the appearance of only a single study to the knowledgebase here is by itself and argument for additional research (and represents a continuing “gap” as identified by the authors).
(9) In the introduction and discussion, an emphasis is presented for young people and older adults. The introduction seems to exaggerate the level of suicide among young people, particularly ages 15-24 with respect to being “more common among.” The rates generally are lower among those 15-24 than among those in adulthood 25-44,45-64, and 65 and older. The “more common in” tag for older adulthood is generally true, particularly those over age 75 and especially men. There are good reasons to focus on young people and suicide, but the characterization as “more common among” in that case might be reconsidered. There is of course much concern about suicide in young ages and increases over time in these age group are certainly a cause for concern and for providing effective interventions, both for those exhibiting suicidal behavior as well as survivors of suicide loss of these ages or among the survivors of suicides in these age groups.
Author Response
Revisor 2
This manuscript presents an overview of grief among suicide loss survivors, with a focus on cognitive behavioral aspects and approaches to suicide grief experiences. The goal of the article is to analyses the research literature on suicide grief, identifying research investigations that have utilized randomized control trials and evaluating the status of the quality and findings of cognitive behavioral therapeutic (CBT) approaches for effectiveness after suicide loss. After extensive search for such research findings, 5 investigations were identified and evaluated. Based on these studies, the authors observed that CBT are effective while the quality of the studies was inadequate and there are limitations to the studies and their methods. Strengths of the study lie in the number of databases included in the search for identifying research on CBT for suicide grief and mourning to include in the review of effectiveness. The use of the PRISMA criteria in identifying the studies is another strength.There are some issues and questions that might be addressed.
(1) The entire paragraph for Section 3.1 Methodological Quality Assessment (lines 295-306) is an exact repetition of the paragraph of 3. Results (lines 265-277). No new information appears in Section 3.1’s text and more important, no information about Methodological Quality Assessment is included, and no information about Figures 2 and 3 appear. Thank you very much for the review. The authors have made the necessary modifications.
(2) The sentence in the introduction, lines 106-107 ends midsentence,. Table 1 is not mentioned in the text and an ending is needed in the sentence. Done. Was there more to add to the Halabi and Garcia table of factors to better understand the inclusion of Table 1? Thank you for the suggestions. To enhance clarity, the authors have removed this table and revised the introduction section.
(3) For Section 3.2 Participant Description, the heterogeneity of the participant relationships is a plus and minus. The plus is a better representation of some of the variety of possible relationships between the deceased and survivors. A minus is that little is known about whether these interventions are effective for some relationship groups but less so or not for others.
We appreciate your comments regarding Section 3.2 and the heterogeneity of participant relationships. We recognize that diversity in relationships presents both advantages and disadvantages. While we agree that this heterogeneity offers a broader representation of the possible dynamics between the deceased and survivors, we also understand the concern about the variable effectiveness of interventions depending on the type of relationship.
In future research, we will consider the possibility of segmenting participant groups based on the nature of their relationship with the deceased in order to more specifically evaluate how these interventions impact different groups. However, we believe that the approach taken in this study provides valuable initial insights and opens the door to more focused research in the future.
Thank you for your valuable feedback.
(4) On line 113 the “popular” use of the term “suicide survivor” is discussed, along with its potential confusion with reference to suicide attempters. However, for some time now the writing and research on the topic of suicide survivors has utilized “suicide loss survivors” for the previously utilized “suicide survivors” terminology alone. At the same time, suicide attempters have often been referred to as “suicide attempt survivors.” The mention of the previous terminology and the confusion it potentially represented is accurate, but at this point in time the issue is minimal and essentially addressed in the language utilized.Thank you for the feedback.The authors have changed the term suicide survivor to suicide loss survivors.
(5) Included in the review in the introduction, the work of Edwin Shneidman (though a secondary source is cited) appears on line 123. It should be noted that Shneidman’s name does not include a “c” (i.e., “Schneidman) as appears in line 123.Thank you for the feedback. The authors have made modifications accordingly.
(6) (a) On line 122 the wording “commit suicide” appears. While this wording has been used in the research and clinical literature in the past (and unfortunately sometimes still is), current practices and recommendations strongly encourage avoidance of such phrasing in labeling deaths by suicide. Instead, wording such as died by suicide, die by suicide, died of suicide, killed themselves, suicide death, death by suicide, or some other wording is suggested. The term “commit suicide” implies, suggests, or is associated with a criminal or immoral act. It is interpreted as a pejorative word and one that is harmful to the surviving family of those who have died by suicide. That implication is associated with feelings of shame and stigmatization toward the individual and often their surviving family members as well. Done
(7) In the Material and Methods section (lines 247 ff) it is noted that a first and second screening were conducted. Who conducted the screenings is not identified here or in the Author Contributions listing at the end of the manuscript. Were the evaluations conducted by all authors, or a subset of authors, research assistants or some other persons?
The selection of relevant studies involved a three-phase process carried out by two reviewers (RMJC & MCG). First, the titles and abstracts of all identified papers were assessed according to the pre-established inclusion and exclusion criteria. In the second phase, the full texts of papers deemed potentially relevant were examined. This phase concluded with a cross-reference check.
(8) The authors contend that the passage of sufficient time has taken place to justify a review of this research topic. The last review is stated to have taken place in 2018. When considering the 5 studies identified and reviewed (no time frame was placed on the search period), only one of the studies (Treml et al., 2021, apparently the “additional record” identified - or was it identified in the searches but another of the other four, though appearing before 2018, represents the “additional record?) was new and identified for inclusion. While this may be the only new study meeting the established criteria, the significance or implication of the lack of other research on this topic after a 6-year period of time is not addressed by the authors. Therefore, while seemingly positive results of effectiveness are stated, the appearance of only a single study to the knowledgebase here is by itself and argument for additional research (and represents a continuing “gap” as identified by the authors).
The authors have taken this into account in the limitations section. The revised text now reads:
"We acknowledge that the limited number of new studies identified since the last review in 2018 suggests a need for further research in this area. Despite the inclusion of one new study (Treml et al., 2021), the absence of additional recent research highlights an ongoing gap that warrants further investigation. This limitation underscores the importance of continued research to build upon the existing knowledge base and address emerging questions in the field."
(9) In the introduction and discussion, an emphasis is presented for young people and older adults. The introduction seems to exaggerate the level of suicide among young people, particularly ages 15-24 with respect to being “more common among.” The rates generally are lower among those 15-24 than among those in adulthood 25-44,45-64, and 65 and older. The “more common in” tag for older adulthood is generally true, particularly those over age 75 and especially men. There are good reasons to focus on young people and suicide, but the characterization as “more common among” in that case might be reconsidered. There is of course much concern about suicide in young ages and increases over time in these age group are certainly a cause for concern and for providing effective interventions, both for those exhibiting suicidal behavior as well as survivors of suicide loss of these ages or among the survivors of suicides in these age groups.
Thank you for the feedback. The authors have made modifications accordingly.
Round 2
Reviewer 2 Report
Comments and Suggestions for Authors
This revised manuscript addressed issues raised in the review process.
There are some minor issues that might be addressed.
(1) Lines 34-36 involve the sentence “Grief among suicide loss 34 survivors is more complex due to existing stigma, affecting both those directly impacted 35 by the suicide attempts and their loved ones, leading to complicated grief.” The grief of suicide loss survivors is directly impacted by the death by suicide, not nonfatal suicide attempts.
(2) Lines beginning with 103: The three paragraphs discussing Boelen et al.’s work could be reconsidered with respect to paragraph breaks. That is, 3 phases are highlighted in the first paragraph wording and only a) and b) appear in that paragraph. There is then a paragraph break after discussing b). The paragraph apparently, it seems, is still discussing b). Later in the new second paragraph "c) Avoidance of grief-related reactions: " appears in the last lines of the paragraph. Then another paragraph seems to be a continuation of the "c)" phase information. Why not start the third paragraph at the “c)” lines of the second paragraph? The authors might want to consider the paragraph switches in this sequence to assist the reader.
(3) Lines 205-209: The text “3. Results. /spanspan This section may be divided by subheadings. It should provide a concise and precise /spanspan >206 /spanspan description of the experimental results, their interpretation, as well as the experimental /spanspan >207 /spanspan conclusions that can be drawn.”/span appears starting on these lines. Then a few lines down “3. Results…” appears. It would seem this text (lines 205-209) should be deleted. It suggests an approach to the organization of the section and sounds more like review comments than content for the article itself./p p class="MsoListParagraphCxSpMiddle" style="margin: 0in 0in 0in 0.5in; caret-color: #000000; color: #000000; font-style: normal; font-variant-caps: normal; font-weight: 400; letter-spacing: normal; orphans: auto; text-align: start; text-indent: 0px; text-transform: none; white-space: normal; widows: auto; word-spacing: 0px; -webkit-text-stroke-width: 0px; text-decoration: none"span style="color: black" /span/p p class="MsoListParagraphCxSpMiddle" style="text-indent: 0pt"(5) Lines 424-425: After “no conflicts of interest.” The text “ure S1: title; Table S1: title; Video 424 S1: title.” Seems to be something to delete.
Author Response
1) Regarding lines 34-36:
Thank you for pointing out the confusion. We have corrected the wording to specifically refer to loss by suicide, rather than suicide attempts. The text now clarifies that grief among suicide loss survivors is more complex due to the stigma associated with death by suicide, which affects both those directly impacted and their loved ones, leading to complicated grief.
2) Regarding the paragraph structure (lines 103):
We appreciate your observation about the organization of the paragraphs. We have reviewed and restructured the sections to make the discussion of Boelen et al.'s three points clearer. Now, the third paragraph starts with point "c) Avoidance of grief-related reactions," ensuring a more coherent and clear transition for the reader.
3) Regarding lines 205-209:
Thank you for this suggestion. We have removed the unnecessary text, which seemed to be more of an internal review comment than part of the manuscript. This change helps make the “Results” section clearer and more precise.
4) Regarding lines 424-425:
We have removed the residual text that was not relevant to this section, including the references to "Figure S1: title; Table S1: title; Video S1: title." This improves the clarity and presentation of the document.
Once again, we greatly appreciate your comments, which have helped improve the clarity and organization of the manuscript.
